# Membrane Protein Stabilization Strategies for Structural and Functional Studies

**DOI:** 10.3390/membranes11020155

**Published:** 2021-02-22

**Authors:** Ekaitz Errasti-Murugarren, Paola Bartoccioni, Manuel Palacín

**Affiliations:** 1Laboratory of Amino acid Transporters and Disease, Institute for Research in Biomedicine (IRB Barcelona), The Barcelona Institute of Science and Technology (BIST), Baldiri Reixac 10, 08028 Barcelona, Spain; paola.bartoccioni@irbbarcelona.org; 2CIBERER (Centro Español en Red de Biomedicina de Enfermedades Raras), 28029 Barcelona, Spain; 3Department of Biochemistry and Molecular Biomedicine, Universitat de Barcelona, 08028 Barcelona, Spain

**Keywords:** membrane proteins, stability, mutagenesis, detergent, lipid, antibody, nanobody, ligand

## Abstract

Accounting for nearly two-thirds of known druggable targets, membrane proteins are highly relevant for cell physiology and pharmacology. In this regard, the structural determination of pharmacologically relevant targets would facilitate the intelligent design of new drugs. The structural biology of membrane proteins is a field experiencing significant growth as a result of the development of new strategies for structure determination. However, membrane protein preparation for structural studies continues to be a limiting step in many cases due to the inherent instability of these molecules in non-native membrane environments. This review describes the approaches that have been developed to improve membrane protein stability. Membrane protein mutagenesis, detergent selection, lipid membrane mimics, antibodies, and ligands are described in this review as approaches to facilitate the production of purified and stable membrane proteins of interest for structural and functional studies.

## 1. Introduction

Membrane proteins are crucial for many physiological processes. They account for about 25% of all proteins encoded by the human genome [1] and about two-thirds of know druggable targets in the cell [2,3], including receptors, channels, and transporters. Membrane proteins are a major pharmaceutical target because they play essential biochemical roles in the transport of molecules across membranes and in cell communication and signal transduction [4]. To design target-directed drugs and shed light on the molecular mechanisms underlying drug activity, it is essential to have structural information about the target protein at atomic resolution. Nevertheless, a large number of these targets have not been structurally solved (only 1201 unique known membrane protein structures have been reported to date) (https://blanco.biomol.uci.edu/mpstruc/, accessed on 18 February 2021), thereby hindering structure-based intelligent drug design. Apart from their low natural abundance and variable toxicity when overexpressed [5,6], the main difficulty is obtaining pure and stable functional membrane proteins [7,8].

Membrane protein expression and purification for structural purposes are challenging. Acceptable levels of protein expression, as well as purified stable protein, are normally prerequisites for any structural technique [7,8]. In this respect, it must be taken into consideration that the lipidic composition of membranes surrounding cells and intracellular compartments plays fundamental structural and functional roles in membrane proteins [9,10,11]. In this context, the direct interactions of human large neutral amino acid transporters LAT1 and LAT2 (L-Amino acid Transporters 1 and 2; SLC7A5 and 8, respectively) with cholesterol are essential for protein stability and function [12,13,14]. Similarly, the interaction of both G protein-coupled receptors (GPCRs) and amyloidogenic peptides with membrane lipids is crucial for protein function and cellular toxicity, respectively [15,16,17,18]. Given that the atomic resolution of membrane proteins requires extraction of the target protein from its native environment, the use of particular detergents and/or lipid combinations is highly relevant for the purification of fully functional membrane proteins [19,20]. However, identifying optimal detergent(s) and buffer conditions for protein stability is often difficult and time-consuming [21,22], although the use of protein-Green Fluorescent Protein (GFP)-fusion constructs facilitates this task [23,24].

Once the target protein has been extracted from the lipidic membrane, it must undergo purification and be stable enough for subsequent protein reconstitution in liposomes for functional studies, crystallization screens, or grid preparation for cryo-EM. In fact, membrane protein instability is indeed the bottleneck for structural and functional studies [22,25]. Protein instability can arise from protein amino acid composition or the presence of multiple conformational or oligomeric states [25]. To overcome stability issues, constructs of target protein orthologues or engineered sequences, including fusion constructs, deletions, and/or single point mutations can be screened [24,26,27]. Nevertheless, selecting the most suitable constructs can be time-consuming. Alternatively, high-throughput screens suitable for the identification of stabilizing molecules such as detergents, lipids, antibodies, and ligands (substrates, inhibitors, agonists, etc.) can facilitate the identification of critical additives for membrane protein stability [21,22,24,28].

This review discusses the strategies to overcome the problems associated with low protein stability. Given the origin of protein instability (amino acid sequence-dependent or conformational flexibility), a variety of strategies, from a practical point of view, are proposed.

## 2. Membrane Protein Mutagenesis

Structural studies of membrane proteins are often hampered by the limited amount of final purified stable and functional protein. Finding general approaches to produce sufficient amounts of polytopic membrane proteins with enough purity and stability for structural studies is a remarkable challenge. In this regard, optimization of the gene sequence encoding the membrane protein target is often necessary. Protein engineering is one of the most widely used and successful strategies for conferring desirable physical chemistry properties to a membrane protein for structural studies [29,30]. In particular, the generation of membrane protein mutant libraries and identification of the best-expressed and most stable versions of these have been widely used [31,32,33]. Sources of mutation include rational design, random mutagenesis, scanning mutagenesis, and consensus mutation.

Protein engineering by point mutations, especially in transmembrane domains, can overcome membrane protein instability [32,34,35]. Given that interactions between transmembrane regions are the major determinants of integral membrane protein folding and insertion [36], side-chain substitutions within these domains are likely to have a greater impact on protein stability [37,38]. Rational design for thermostabilizing mutants could help to improve the identification of successful mutants. However, the low amount of high-resolution structural information about membrane proteins and the molecular mechanisms underlying the stabilization effect of a point mutation are often difficult to interpret [39] even after solving the 3D structure, thereby hampering the formulation of general rules for future predictions.

### Systematic vs. Random Mutagenesis Approach

In this scenario, one of the most efficient approaches to optimize selected targets for structural studies consists of screening large libraries of mutants generated either by systematic [40,41,42] or random mutagenesis [32,43]. Systematic mutagenesis, usually by alanine or leucine scanning, has been successfully applied in the field of membrane protein thermostabilization, enabling the purification and crystallization of several challenging targets [44,45]. In this approach, residues in a target protein are systematically substituted for alanine at selected positions by site-directed mutagenesis, thereby allowing the identification of positions that are important for protein function and stability [46,47,48]. However, this approach involves the production of many mutant proteins—a process that is laborious and time-consuming. In contrast, the consensus mutagenesis approach is based on multiple sequence alignments of homologous proteins that identify amino acids that tend to be more prevalent at any given position in a protein family [49,50]. Given this observation, the use of site-directed mutagenesis to introduce the most common amino acids at selected positions frequently leads to stabilized protein variants [51]. In this regard, recent reports have described the application of consensus mutagenesis to membrane proteins [42,52,53].

Finally, random mutagenesis consists of introducing random mutations into the protein gene. The most commonly used random mutagenesis method is error-prone Polymerase Chain Reaction (PCR), which introduces random mutations during PCR by reducing the fidelity of DNA polymerase [54]. Random mutagenesis is a simple method that has been successfully used in membrane protein stability screens [32,43,53]. However, the large numbers of mutants to be analyzed hampers this approach.

Thus, the feasibility of these approaches relies on high-throughput methods to assess the functionality, subcellular localization, and stability of the mutants [21,22,32,55]. In this regard, combining mutagenesis with the use of a fluorescent protein (FP) as a fusion tag has contributed to speeding up this task [22,24]. However, FP tagging can interfere with crucial protein parameters, such as protein activity, complex formation, and subcellular localization, among others [56,57,58]. The use of split-GFP technology overcomes possible issues associated with an FP-tagged protein of interest. Cabantous and Waldo reported the possibility of splitting the eleven β-strands of GFP into two non-fluorescent fragments (strands 1–10 and strand 11), which could be spontaneously reassembled and become fluorescent (Figure 1) [59]. Thus, by fusing only a small portion of the GFP (strand 11) to a membrane protein, any eventual impact of the full-length GFP during translation and insertion of the membrane protein is minimized. Therefore, sequential co-expression of the membrane protein mutants tagged to strand 11 (GFP11), followed by GFP1–10, results in fluorescence emission when the two GFP fragments complement each other (Figure 1) [59,60,61]. This process occurs only when a particular mutant is properly folded and inserted into the cell membrane [32] and can rapidly identify misfolded or poorly stable proteins as a result of mutagenesis.

## 3. Detergent Selection

The environmental complexity of the lipid bilayer usually leads researchers to transfer the target membrane protein to a more controlled environment for experimental study [19,20]. This synthetic system normally consists of a solubilizing amphiphilic molecule, which shields the transmembrane hydrophobic core and brings extracellular regions into contact with an aqueous phase. Detergents are the molecules of choice as they have unique properties in aqueous solutions, creating a mimic of the natural bilayer [19,20]. Nevertheless, the detergent micelle is not usually able to maintain membrane protein function and structural stability to the same extent as the native membranes [62,63,64]. Hence, selection of a suitable detergent is one of the main bottlenecks in structural and functional studies of membrane proteins.

Given the huge variety of detergents currently available, the identification of a detergent (or a group of detergents) that fulfills specific protein requirements, in terms of stability and functionality, can be a painstaking trial-and-error process [19,20,22]. N- or C-terminal fluorescent-tagged membrane proteins facilitate monitoring of the quality of the expressed fusion protein by whole-cell detergent solubilization, followed by fluorescence size exclusion chromatography analysis (FSEC), prior to purification [23,24]. Additionally, stability in detergents has been studied using a variety of techniques, including analytical size exclusion chromatography [23,65], light-scattering-based techniques [21,66,67], sedimentation equilibrium centrifugation [67], differential filtration and ultracentrifugation assays [68], and thermal denaturation assay [21,66,69]. Nevertheless, the choice of an appropriate detergent is difficult and is also influenced by the task to be carried out and by the intrinsic stability of the target protein, in addition to the properties of the detergent itself.

### Classification of Detergents

Detergents are classified into three major categories on the basis of their structure and ionic properties (non-ionic, ionic, and zwitterionic). Additionally, they are also categorized as “mild” or “harsh” as a function of their tendency to maintain protein stability.

Non-ionic detergents contain uncharged hydrophilic glucose-derived or polyoxyethylene head groups [19,20,70]. This type of detergent is considered to be mild as the destabilization they cause is almost completely reversible [71,72,73]. This capacity is explained by the fact that non-ionic detergents preferentially break lipid–lipid and lipid–protein interactions, thereby supporting intramolecular helix–helix interactions and conserving key structural features [73]. However, the degree of destabilization correlates well with the length of the acyl chain of the detergent. In this regard, short (C7–C10) hydrocarbon chains (e.g., octylglucoside and nonylmaltoside) can often lead to deactivation of the protein, in contrast to their corresponding intermediate (C12–C14) chain-length derivatives (e.g., dodecylmaltoside) [74].

Ionic detergents are composed of a charged head group (cationic or anionic) and a hydrophobic acyl chain or steroidal backbone, the critical micellar concentration (cmc) being dependent on the concentration of counter-ions [19,20,70]. This type of detergent (particularly those containing acyl chains) is considered “harsher” than uncharged detergents as they are highly effective at solubilizing membrane proteins but almost always denature them. Ionic detergents disrupt mainly protein–protein and intraprotein interactions directly, and not lipid–protein interactions [73,75]. Steroid-based detergents such as bile acid salts are milder, leading to less inactivation than hydrocarbon detergents containing the same head group [74,76]. The use of ionic detergents in membrane protein biology has been limited mostly to protein reconstitution into lipidic systems, and these detergents have often been used unsuccessfully in the field of structural biology.

Zwitterionic detergents represent an intermediate step between ionic and non-ionic detergents, combining the properties of these two detergent groups [19,20,70]. Although, in general, more deactivating than non-ionic detergents, zwitterionic detergents have been successfully used in the atomic structure resolution of several membrane proteins [77]. In particular, *N*,*N*-dimethyldodecylamine N-oxide (LDAO), a small micelle-forming zwitterionic detergent, leads to tighter packing in the crystal lattice and thus enhanced diffraction [22,77]. Nevertheless, small micelles often result in protein destabilization and are useful only for a small subset of membrane proteins. Additionally, as the head group charge depends on pH, solubilization and purification conditions may modify detergent properties [70], which may complicate purification setup.

As a rule of a thumb, larger micelles, like those formed by long acyl chain detergents, can easily accommodate the hydrophobic core of the protein, thereby favoring the retention of functional and structural properties [78]. However, large micelles can hinder crystal contacts, thereby producing lower resolution structures [22,77], which is particularly relevant for plasma membrane transporters that often lack large hydrophilic domains. Although shorter detergents would result in higher resolution crystals, a more stable protein in longer detergents would be of interest for activity tests or reconstitution into lipidic systems such as liposomes and nanodiscs.

Given the above considerations and despite the wide commercial availability and the continuous development of detergents with diverse chemical structures and physical properties, only a few are of general use for membrane proteins. However, the development of new tools in the field of structural biology has allowed the use of detergents that have been regularly discarded for such purposes. A recent example of this is digitonin, which is used as a detergent for Cryo-EM structural resolution of membrane transporters [79,80,81].

## 4. Lipid Membrane Mimics

Certain membrane proteins show a high propensity to aggregate during isolation from their native lipidic environment. In fact, detergent micelles do not fully resemble the native environment as they remove protein–lipid interactions that could be key for protein function and/or structure [82,83]. Additionally, the removal of specific lipids during purification—a process called delipidation—can severely impair protein stability and function [83,84,85]. The structural biology of membrane proteins provides insight into the roles of specific membrane lipids within the protein–lipid environment. In this regard, recently solved neutral amino acid transporter hLAT1/CD98hc showed the presence of a cholesterol molecule that supports transmembrane domain connections between the light and heavy subunits, and thus, stabilizing the complex [86]. This cholesterol-binding site seems to be conserved across other members of the SLC7 family, suggesting a specific role for this protein–lipid interaction in heterodimer stabilization and activity [12,13,14]. Moreover, membrane lipids induce lateral pressure, which has a stabilizing influence on membrane proteins [87,88]. Extraction and purification of membrane proteins in lipid/detergent micelles, where the hydrophobic core of the membrane protein is solvated with the nonpolar lipid groups, can overcome this problem [83,89].

Reconstitution into liposomes is a widely used strategy for the functional characterization of membrane proteins [13,14,35,86]. In fact, lipidic composition of liposomes can be fine-tuned, identifying specific lipidic requirements of the target protein for proper function [13,14]. Additionally, lipid membrane mimics, such as lipidic cubic phase (LCP), nanodiscs, and styrene maleic acid co-polymer lipid particles (SMALPs), have been developed and allow for the stabilization of membrane proteins in a controlled, membrane-mimicking environment for functional and structural studies [90,91,92].

The LCP is a liquid crystalline material, a mesophase, consisting of a principal lipid (typically monoacylglycerol), which adopts the form of a lipid bilayer [90]. The principal lipid is usually mixed with additive lipids (e.g., cholesterol) to give the desirable physical, chemical, and functional properties to the lipidic environment. LCP constitutes a membrane mimetic matrix that promotes solubilization, stabilization, and crystallization of specific membrane proteins [34]. In this line, using LCP for membrane protein structural studies requires protein reconstitution into the lipid bilayer, which happens spontaneously as the lipid and detergent-solubilized protein solutions are mixed to homogeneity forming the mesophase. However, despite recent advances in techniques, tools, and materials, obtaining a well-formed mesophase that includes the protein of interest remains a challenging task.

Developed by Sligar and colleagues in 2007, nanodisc technology is based on an engineered form of the human high-density lipoprotein fraction, which acts as a membrane scaffold protein (MSP) for nanodisc assembly [93]. In this regard, in the presence of lipids, two MSP molecules can self-assemble into a discoidal phospholipid bilayer in a belt-like way to form a nanodisc (Figure 2). The incorporation of an integral membrane protein into a nanodisc bilayer is controlled by the incubation of the detergent-solubilized protein with a mixture of MSP and lipid in a stoichiometric-dependent manner (Figure 2) [93,94]. By fine-tuning the ratio of these constituents, proper nanodisc assembly can be initiated after detergent removal. Once the nanodisc has been assembled, the intracellular and extracellular domains of the membrane protein are exposed to solvent while the hydrophobic protein core remains protected by the lipid acyl chains, which in turn interact with the amphipathic helices of the MSP (Figure 2). Furthermore, nanodisc diameter can be easily controlled by the length of the MSP used (Table 1), thereby fitting a wide variety of membrane proteins, including protein complexes [94,95].

### Nanodisc Applications in Membrane Protein Biophysics

The lipidic environment modulates membrane protein functionality [12,82,83] by molecular interactions between the protein and lipids of the phospholipid bilayer. To answer key questions regarding the requirement of specific phospholipids for optimal activity or stabilization of particular conformations, nanodisc technology provides a controllable membrane environment by regulating the lipid composition with precision [96]. However, simulation of the cellular native environment requires the selection of an appropriate lipid composition for nanodisc assembly. Additionally, the phase transition temperature (Tm) of the lipids used must be taken into account. In this regard, experiments conducted with nanodiscs, including nanodisc assembly, must be performed above the Tm of the lipid(s) used to ensure that the lipid bilayer remains in the native liquid crystalline phase [97].

Nanodiscs are also an excellent tool through which to study protein complexes and protein–protein interactions. To this end, co-reconstitution of multiple proteins into a single nanodisc can lead to the formation of protein complexes in a controlled membrane environment [98,99]. However, multiple purification steps and protein conformational heterogeneity, together with the laborious optimization of the proteins to MSP ratios and the randomized nature of protein incorporation into nanodiscs, often lead to poor complex formation yields. Additionally, as also reported for membrane protein liposome reconstitution [100,101], protein integration into the nanodisc can be bi-directional, thereby limiting the formation of native protein complexes.

Protein–lipid and protein–protein interaction studies in nanodiscs have been regularly performed with detergent-solubilized and purified proteins. This process often destabilizes membrane proteins as they are removed from the native membrane and inserted into a non-native membrane environment. Additionally, functionally critical protein–protein and protein–lipid interactions may be lost during protein purification [83,84,85]. Finally, the asymmetry in lipid composition between the inner and the outer leaflets observed in cell membranes is difficult to reproduce in nanodiscs [102].

To solve these issues, nanodisc technology has been extended to directly capture membrane proteins and protein complexes from the native environment of cellular membranes [103]. Thus, a native membrane preparation can be detergent-solubilized in the presence of excess MSP and lipid, whereupon detergent removal would result in a nanodisc library containing the starting membrane protein population [103]. This system removes protein overexpression and purification setups, capturing poorly expressed proteins or intact protein complexes and maintaining the native protein–lipid interactions. However, these MSP nanodiscs require prior detergent solubilization of the native membranes necessary for subsequent insertion of the solubilized membrane patches into the nanodisc. An alternative detergent-free approach takes advantage of the properties of the styrene maleic acid (SMA) co-polymer scaffolding component [92]. The amphipathic nature of SMA (hydrophobic styrene and hydrophilic maleic acid) allows self-insertion into biological membranes and extracts small discs of lipid bilayer, encircled by polymer and named SMA lipid particles (SMALPs). Such SMALPs can be further purified to isolate the lipid particle in which the protein of interest is embedded [92].

Finally, nanodiscs containing membrane protein are a powerful tool for the generation of antibodies against the target protein. In fact, phage display with nanodiscs is effective for the structural characterization of membrane proteins [104,105]. The use of detergents for protein extraction can result in target destabilization and the loss of conformational epitopes. Thus, the incorporation of membrane proteins into nanodiscs favors native protein folding, thereby facilitating the identification of antibodies against the selected target.

## 5. Antibodies

Among the aforementioned membrane protein stabilization strategies, antibodies against a particular membrane protein provide the most rational target-directed stabilization approach for further functional and structural studies. The structures of many membrane proteins have been solved without the aid of antibodies; however, the presence of detergent micelles can impede protein–protein interactions, which are critical for the formation of an ordered crystal lattice [101]. In this regard, monoclonal antibody fragments can lead to successful 3D structure determination, especially for membrane proteins that form low resolution diffracting crystals [106,107]. In fact, Fab or Fv fragments derived from monoclonal antibodies have been used as a powerful chaperone, thus allowing new structures to be solved for a variety of membrane proteins, including channels, receptors, and membrane transporters [107,108,109]. Antibody fragments facilitate membrane protein crystallization by increasing the number of protein–protein interactions within the crystal lattice [101]. Unfortunately, most commercial monoclonal antibodies rarely favor crystal contacts as they have been raised against peptide fragments from the target proteins. Therefore, antibody binding to these lineal epitopes increases protein flexibility, thus hindering crystal formation. In this respect, most of the antibodies successfully used in the elucidation of the 3D structure of antibody–membrane protein complexes recognize conformational rather than lineal epitopes [107,108,109]. Nevertheless, the generation of useful conformational monoclonal antibodies is generally challenging, time-consuming, and expensive, although several methodological attempts to overcome these issues have been made [110]. Additionally, such antibodies are not necessarily stable in the context of a reducing environment like that of the cellular cytoplasm. In this regard, nanobody technology based on camelid heavy chain-only antibodies (HcAbs) has overcome many of these shortcomings [111].

### 5.1. Nanobodies

HcAbs are 95–96 kDa unique IgGs found in the serum of camelids (dromedaries and llamas) and cartilaginous fishes (sharks and rays) [112,113]. They are composed of two heavy chains lacking the light chain and the first constant domain (Figure 3a). The HcAb antigen-binding domain, which is composed of a single variable domain named VHH or nanobody, has a molecular weight of 12–15 kDa and harbors the full antigen-binding capacity of the parental antibody [113]. The nanobody has three complementary determining regions (CDR1–3), which form the antibody paratope (Figure 3b). These regions are organized in loops and are surrounded by four more conserved framework regions (FR1–4) [113]. CDRs are highly diverse in sequence and are responsible for antigen binding. To provide a sufficient surface for antigen interaction, CDRs (especially CDR1 and CDR3) are longer than those of conventional antibodies and are linked to each other by disulfide bridges to restrict flexibility, thereby ensuring high-affinity antigen binding (Figure 3b) [113]. Due to their small size and compact shape, nanobodies feature a convex paratope, which has access to cavities or clefts on the surface of proteins that are often inaccessible to conventional antibodies [114]. Additionally, camelids immunized with properly folded membrane proteins produce antibodies against conformational epitopes, which are composed of amino acid segments clustered only within the native protein (Figure 3c) [101,115]. Upon induction of a sufficient immune response, antigen-specific nanobodies are obtained by cloning the variable VHH gene population from peripheral blood lymphocytes. A final selection of nanobodies against the target protein is then made using one of the many combinatorial biology methods available, including phage display, yeast display, and ribosome display [116].

One of the most interesting features of nanobodies is their enhanced solubility and stability compared to conventional antibodies [113]. The molecular bases underlying these features reside in the composition of FR2, which contains a number of hydrophobic amino acids that mediate the interaction with the variable domain of the light chain. As nanobodies lack the light chain, these residues have been substituted by hydrophilic amino acids, thereby reducing the likelihood of their aggregation and increasing their solubility [113]. Additionally, the small size of nanobodies, together with the fact that they are encoded by single gene fragments, provides many advantages, one of which is that they can be easily produced recombinantly at high levels in a variety of expression systems, including microorganisms [111,117,118].

### 5.2. Nanobody Applications in the Functional and Structural Biology of Membrane Proteins

Nanobodies can be easily customized to add a particular tag (e.g., fluorescence, affinity, epitope tag, etc.) without the loss of affinity or stability [119]. This is the case of chromobodies, which are obtained by fusing nanobodies to fluorescent proteins (FPs) used to image the localization of the target protein within cells or to quantify expression [120]. The use of chromobodies overcomes possible issues associated with the use of an FP-tagged protein of interest. On the other hand, adding nanobodies onto particular protein scaffolds results in increased molecular weight nanobody derivatives, named megabodies. These molecules have been used, among other applications, to obtain 3D reconstructions of membrane proteins that are too small to allow accurate particle alignment by Cryo-EM [121]. Nevertheless, the most straightforward application of nanobodies in structural biology is associated with the improvement of crystal diffraction quality [101,115]. Like conventional antibodies, nanobodies can also rigidify flexible regions and also mediate protein–protein contacts within the crystal lattice, thereby contributing to high-quality crystal packing [101]. Additionally, as they are normally raised against native and properly folded membrane proteins, nanobodies can target specific conformations of the translocation cycle [115,122]. In fact, the small size of nanobodies, together with the flexibility of CDRs, makes them useful tools for stabilizing different conformational states in flexible proteins [123]. In this regard, high-energy, low-population, or unstable conformers can be selected and thus, functionally and structurally characterized [115,123]. On the other hand, nanobodies that bind discontinuous epitopes that span more than one protein have also been used to crystallize transient multiprotein assemblies, which are more rigid in complex with a nanobody and thus, provide a better starting point for the crystallization of unstable membrane protein complexes [124]. Finally, nanobodies can also be used to characterize different substrate binding sites on the same protein by selectively inhibiting them. This is the case of the bacterial L-amino acid transporter BasC, where the use of nanobodies allowed the unequivocal functional characterization of the extra and intracellular binding sites, which present different affinities for their substrates [101].

## 6. Ligands

Membrane proteins exert key cellular functions. Most of these are regulated by ligand binding with various modes of action. Substrates, inhibitors, agonists, antagonists, proteins necessary for the formation of protein complexes, lipids, and detergents, among others, could be included within this category. Membrane proteins are particularly challenging due to their inherent flexibility and instability in detergents during extraction and purification [22,25]. Thus, identification of suitable purification conditions that can help to stabilize functional proteins is critical for subsequent biophysical studies [22,77]. In this regard, membrane protein binding to small ligands is of particular interest. It is generally accepted that small-molecule binding induces protein conformational changes, which, in turn, may alter protein flexibility. In this regard, the interaction of proteins with small ligands often results in increased protein thermostability [101,125], which has been associated with a substantial increase in structural order and protein packing [126]. However, some ligands destabilize proteins by binding primarily to the unfolded state of the protein and destabilizing it, thereby reducing protein thermostability [126]. Therefore, systematic identification of ligands that stabilize the target protein might significantly improve the success rates of protein purification, structural determination, and functional characterization [22,77,125].

The identification of membrane protein ligands is still a bottleneck for functional and structural studies. In this regard, a variety of techniques allowing high-throughput ligand analysis have been reported in the last decade. Typically, the most direct way to identify ligands for a particular membrane protein is by using binding techniques. These include the scintillation proximity assay (SPA), which allows effective identification of ligands by direct binding to the protein of interest or by inhibition of a known target protein substrate [55,127]. Despite the advantages of this technique, the need for a radioactive ligand or substrate and high affinity of the protein for the radiolabeled molecule, in addition to the need for purified protein, are aspects that limit the applicability of this type of assay in the search for ligands.

In this context, the use of GFP-based thermal stabilization technologies to identify protein ligands became popular a decade ago [22,125]. The simplicity of these approaches and the possibility to employ them with solubilized, unpurified protein led to their use being extended to a large number of membrane proteins [22,125]. However, FSEC analysis slows down the screening process and limits the number of ligands that can be analyzed. Alternatively, a more high-throughput analysis such as microscale thermophoresis (MST) is effective at identifying ligands for non-purified GFP-tagged membrane proteins [128]. Nevertheless, GFP fusion constructs may result in altered target protein stability. An alternative is to study thermostabilization by analyzing the intrinsic fluorescence of the protein [129]. However, this technique requires the use of purified protein, thereby limiting its use. Another option focuses on the use of fluorescent dyes, such as the thiol-specific fluorophore N-[4-(7-diethylamino-4-methyl-3-coumarinyl)phenyl]maleimide (CPM) or SY PRO-orange, to study thermic unfolding processes in the presence of ligands [69,130]. However, the hydrophobic nature of reporter dyes results in their interaction with solvent-accessible hydrophobic areas of membrane proteins, as well as with detergent micelles and molecules, thereby causing increased background and hindering analysis of the results [130]. Finally, label-free methods based on light scattering are also effective for assessing the thermostability of membrane proteins by ligand binding, even in the presence of detergents [131].

## 7. Conclusions

Selection of the best membrane protein sequence could be a key step to solving the structure of unstable membrane proteins. Systematic or random mutagenesis can improve the stability of a given protein of interest.

Selection of the optimal combination of detergent and lipids is mandatory to achieve a stable purified protein. A GFP-tagged version of the protein to speed-up detergent screening is, at present, a common strategy. The addition of specific lipids or reconstitution of the membrane protein in lipid membrane mimics can improve the stability of the membrane protein. These membrane mimics provide a more native environment than detergent micelles.

The use of ligands (particularly small molecule binders or antibodies) can improve protein stability and/or freeze the protein of interest in a particular conformation. Moreover, antibodies and, more particularly, nanobodies are powerful tools to fix conformations and eventually increase protein contacts for crystallization. Recently, the enlargement of nanobodies by fusion with a scaffold protein to generate megabodies facilitated cryo-EM studies for small and difficult proteins.

## Figures and Tables

**Figure 1 membranes-11-00155-f001:**
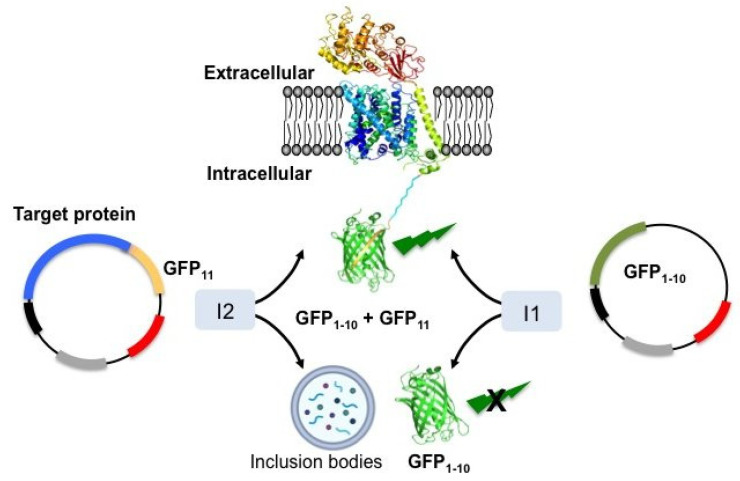
Schematic representation of the split GFP assay as reporter of membrane protein expression and stability in the membrane. The split GFP assay consists of two plasmids: one containing the target membrane protein fused to GFP strand 11 (**left**) and another plasmid expressing GFP strands 1 to 10 (**right**). Protein expression is controlled by two promoters activated by different inducers (I1 and I2). Inducing these two genes sequentially allows the identification of target membrane protein variants that are expressed and inserted into the plasma membrane of the expression system, since the two fragments of the GFP complements resulting in fluorescence emission. Variants confined into inclusion bodies show no fluorescence as no complementation occurs.

**Figure 2 membranes-11-00155-f002:**
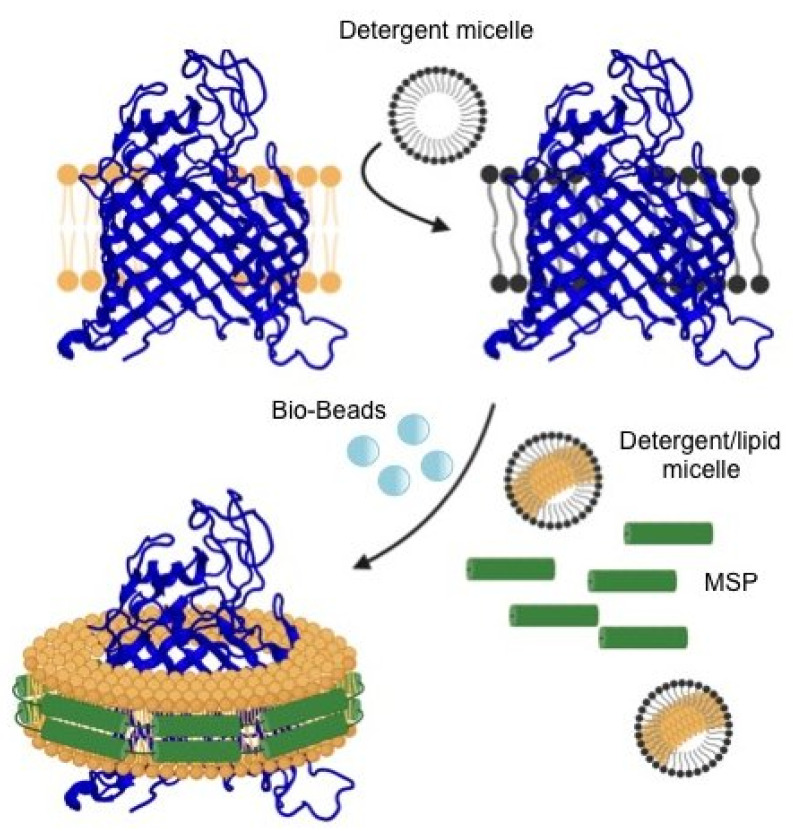
Membrane protein incorporation into a nanodisc. Detergent (black micelle)-solubilized membrane protein (blue) is incubated with a mixture of membrane scaffold protein (MSP; green) and detergent-solubilized lipids (tan). Nanodisc assembly is initiated after detergent removal.

**Figure 3 membranes-11-00155-f003:**
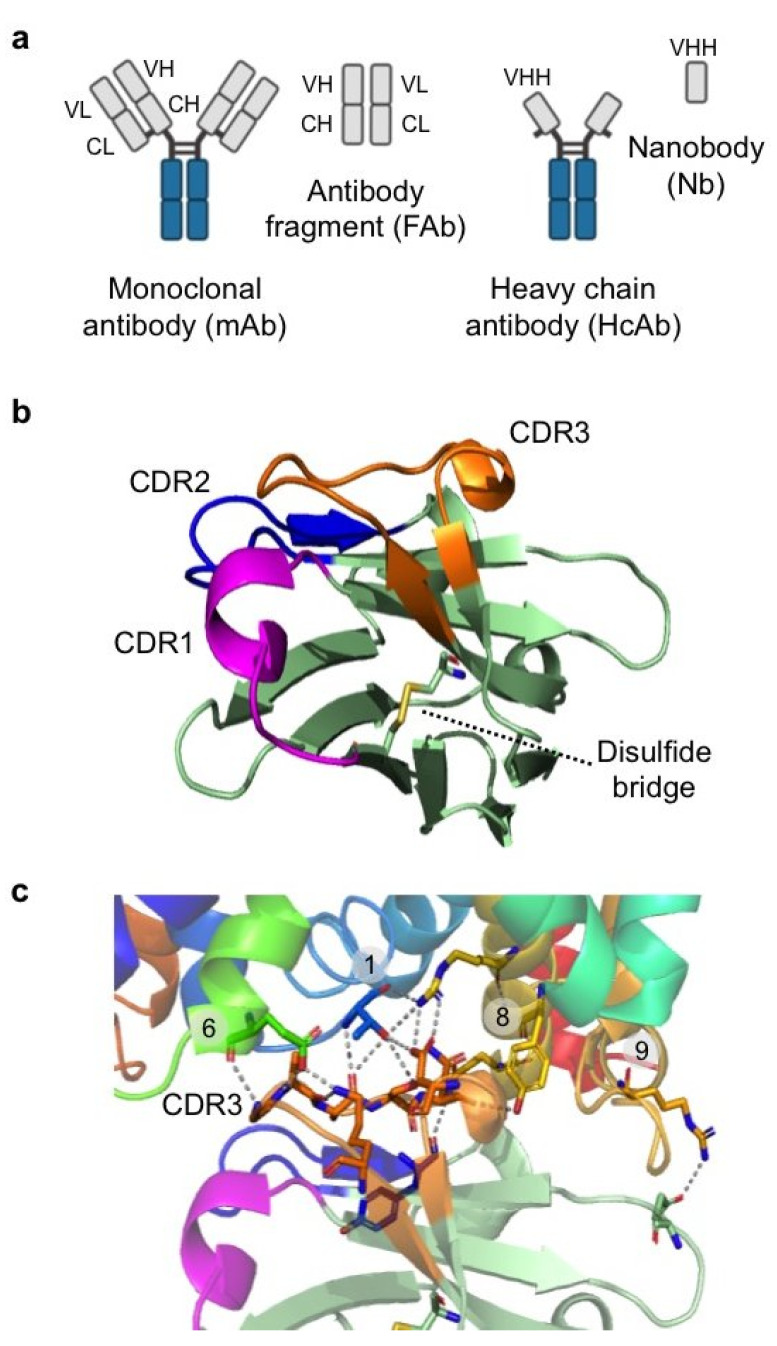
Nanobodies in membrane protein structural biology. (**a**) Traditional monoclonal antibodies (mAb) and their fragments (Fab) vs. heavy chain antibodies (HcAb) and nanobodies (Nb). (**b**) Nanobody structure (PDB ID: 6f2g) [101], showing complementary determining regions (CDR) 1 (magenta), 2 (blue), and 3 (orange) and the disulfide bridge established between Cys 22 and Cys 96. (**c**) Extensive interactions were found between the side chains and backbones of the nanobody CDR3 (orange) and the bacterial amino acid transporter BasC (PDB ID: 6f2g) [101]. CDR3 interacts with residues from BasC transmembrane domains (TMs) 1, 6, 8 and 9.

**Table 1 membranes-11-00155-t001:** Membrane scaffold proteins. A summary of the most used MSPs in nanodisc formation including final nanodisc diameter, molecular mass, and particular features associated with each MSP is shown.

Membrane Scaffold Protein (MSP)	Nanodisc Diameter (nm)	Features	Molecular Mass (Da)
MSP1	9.7	Original MSP1 (deletion 1–43 mutant of human Apo A-1)	24,608
MSP1TEV	9.7	MSP1 with removable 7-his tag	26,930
MSP1D1	9.7	Deletion 1–11 mutant of MSP1TEV	24,662
MSP1D1 D73C	9.6	D73C MSP1D1 mutant in helix 2, Apo A-1 numbering	24,650
MSP1D1 (-)	9.6	MSP1D1 lacking 7-His tag	22,044
MSP1D2	9.6	MSP1 variant lacking the first helix.	24,608
MSP1E1	10.4–10.6	Extended MSP1, helix 4 repeated	27,494
MSP1E2	11.1–11.9	Extended MSP1, helices 4 and 5 repeated	30,049
MSP1E3	12.1–12.9	Extended MSP1, helices 4, 5, and 6 repeated	32,546
MSP1E1D1	10.5	Extended MSP1D1, helix 4 repeated	27,547
MSP1E2D1	11.1	Extended MSP1D1, helices 4 and 5 repeated	30,103
MSP1E3D1	12.1	Extended MSP1D1, helices 4, 5, and 6 repeated	32,600
MSP2	9.5	Fusion of two MSP1 with GT-linker	48,020
MSP2N2	15.0–16.5	Fusion of MSP1D1 and MSP1D2 with GT linker	45,541
MSP2N3	15.2–17	Fusion of MSP1D1 and MSP1D1-17 (deletion amino acids 1–17) with GT linker	46,125
MSP1FC	9.7	MSP1D1 with C-terminal FLAG-tag	25,714
MSP1FN	9.6	MSP1D1 with N-terminal FLAG-tag	25,714

## Data Availability

Not applicable.

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
