# Peer review of "Membrane Protein Stabilization Strategies for Structural and Functional Studies"

_membranes, 2021, doi:10.3390/membranes11020155_

Round 1

Reviewer 1 Report

This manuscript reviewed comprehensively the strategies to stabilize the membrane proteins for structural and functional studies of the membrane proteins. The contents are well organized, and the scientific contents are sound. I suggest the publication of this manuscript after some modifications of the expressions. The detailed comments are as follows.

1.In Line 95, 161, and 299, the use of the subsections is clearly incomplete (with only one subsection or the numbering is wrong). Please check and revise accordingly.

2. For the Section 2 and 3, I suggest adding some figures about the concerned molecules or strategies, which would be helpful for readers. This is a suggestion, not a requirement. Please judge based on your taste.

3. In the last paragraph of Section 6, there are too many "however". It is a little bit colloquial. I suggest reorganizing the contents for clarity.

Author Response

Comments and Suggestions for Authors

This manuscript reviewed comprehensively the strategies to stabilize the membrane proteins for structural and functional studies of the membrane proteins. The contents are well organized, and the scientific contents are sound. I suggest the publication of this manuscript after some modifications of the expressions. The detailed comments are as follows.

1.In Line 95, 161, and 299, the use of the subsections is clearly incomplete (with only one subsection or the numbering is wrong). Please check and revise accordingly.

Checked and corrected as proposed by reviewer.

  1. For the Section 2 and 3, I suggest adding some figures about the concerned molecules or strategies, which would be helpful for readers. This is a suggestion, not a requirement. Please judge based on your taste.

A new figure (Figure 1) showing split GFP complementation protocol has been added as suggested by the reviewer.

  1. In the last paragraph of Section 6, there are too many "however". It is a little bit colloquial. I suggest reorganizing the contents for clarity.

Text has been re-sent to the English editor and suggested paragraph modified.

Reviewer 2 Report

The manuscript by Errasti-Murugarren et al. clearly summarizes the most recent strategies for producing membrane proteins in a form suitable for structural studies. Some amendments will further improve and complete the manuscript:

A mention to lipidic cubic phase is lacking and should be added citing the reference doi:10.1038/nature14655

Line 47. A recent study on cholesterol effect on LAT1 needs to be cited: doi: 10.1038/s41598-020-73757-y

In section 2 mutagenesis add citation doi:10.1038/nature13306

Line 162 only three types of detergents are described.

Liposome reconstitution is only cited; it should be addressed.

Author Response

Comments and Suggestions for Authors

The manuscript by Errasti-Murugarren et al. clearly summarizes the most recent strategies for producing membrane proteins in a form suitable for structural studies. Some amendments will further improve and complete the manuscript:

A mention to lipidic cubic phase is lacking and should be added citing the reference doi:10.1038/nature14655

A mention to lipidic cubic phase has been added in the Lipid membrane mimic section (Section 4, 3rd paragraph, page 5), adding the suggested reference (reference 34).

Line 47. A recent study on cholesterol effect on LAT1 needs to be cited: doi: 10.1038/s41598-020-73757-y

The suggested reference has been added (reference 14) and included in the Introduction and Lipid membrane mimic sections (section 1 and 4, pages 2 and 5).

In section 2 mutagenesis add citation doi:10.1038/nature13306

The suggested reference has been added (reference 34) and included in the Membrane protein mutagenesis section (section 2, page 2).

Line 162 only three types of detergents are described.

Changed as suggested. Thanks for the observation.

Liposome reconstitution is only cited; it should be addressed.

Liposome reconstitution has been defined and cited in Lipid membrane mimic section (section 4 pages 5, 2nd paragraph).